# Structures of RecBCD in complex with phage-encoded inhibitor proteins reveal distinctive strategies for evasion of a bacterial immunity hub

Martin Wilkinson[1†], Oliver J Wilkinson[2], Connie Feyerherm[2], Emma E Fletcher[2‡], Dale B Wigley[1]*, Mark S Dillingham[2]*

[1]Section of Structural Biology, Department of Infectious Disease, Faculty of Medicine, Imperial College London, London, United Kingdom; [2]DNA:protein Interactions Unit, School of Biochemistry, University of Bristol, Bristol, United Kingdom

*For correspondence:
d.wigley@imperial.ac.uk (DBW);
mark.dillingham@bristol.ac.uk
(MSD)

Present address: †School of
Molecular and Cellular Biology,
University of Leeds, Leeds,
United Kingdom; ‡Division
of Protein and Nucleic Acid
Chemistry, MRC Laboratory of
Molecular Biology, Cambridge,
United Kingdom

Competing interest: The authors
declare that no competing
interests exist.

Reviewing Editor: Edward H
Egelman, University of Virginia,
United States

**Abstract** Following infection of bacterial cells, bacteriophage modulate double-stranded DNA break repair pathways to protect themselves from host immunity systems and prioritise their own recombinases. Here, we present biochemical and structural analysis of two phage proteins, gp5.9 and Abc2, which target the DNA break resection complex RecBCD. These exemplify two contrasting mechanisms for control of DNA break repair in which the RecBCD complex is either inhibited or co-opted for the benefit of the invading phage. Gp5.9 completely inhibits RecBCD by preventing it from binding to DNA. The RecBCD-gp5.9 structure shows that gp5.9 acts by substrate mimicry, binding predominantly to the RecB arm domain and competing sterically for the DNA binding site. Gp5.9 adopts a parallel coiled-coil architecture that is unprecedented for a natural DNA mimic protein. In contrast, binding of Abc2 does not substantially affect the biochemical activities of isolated RecBCD. The RecBCD-Abc2 structure shows that Abc2 binds to the Chi-recognition domains of the RecC subunit in a position that might enable it to mediate the loading of phage recombinases onto its single-stranded DNA products.

## Editor's evaluation

This important study addresses the ways in which bacteriophages antagonise or coopt the DNA restriction and/or recombination functions of the bacterial RecBCD helicase-nuclease. The evidence from both biochemistry and structural biology showing convergent evolution is convincing.

## Introduction

The RecBCD complex of *Escherichia coli* plays a dual role in bacterial cell biology, acting both as a phage immunity hub and as the primary mechanism to initiate the recombinational repair of double-stranded DNA breaks (DSBs) (*Dillingham and Kowalczykowski, 2008*; *Levy et al., 2015*; *Millman et al., 2020*). This is facilitated by RecBCD's ability to switch between different modes of action at free DNA ends. In the destructive mode, RecBCD recognises free DNA ends and acts as a processive nuclease that degrades both strands of the DNA duplex into fragments of single-stranded DNA. This can be detrimental to bacteriophage for multiple reasons (for details, see *Dillingham and Kowalczykowski, 2008*; *Levy et al., 2015*; *Bobay et al., 2013*); it can directly degrade and inactivate linear phage genomes upon entry into the bacterial cell, provides immunity against future phage infections because the DNA fragments are captured by Cas1-Cas2 integrase for CRISPR libraries,

and prevents phage replication in the rolling circle mode. However, RecBCD can also act in a recombinogenic mode to convert DSBs into substrates for RecA-dependent strand exchange and recombination with a homologous undamaged DNA molecule. This facilitates DNA break repair which also underpins replication and protects the genetic stability of the host genome. The switch from the destructive to the recombinogenic mode is triggered during RecBCD translocation along DNA by recognition of a specific octameric ssDNA sequence called Chi (crossover hotspot instigator; 5'-GCTG GTGG-3') (*Cheng et al., 2020*; *Bianco and Kowalczykowski, 1997*). Chi recognition causes profound biochemical changes in RecBCD. The enzyme continues to translocate and unwind DNA beyond Chi, upregulates cleavage on the 5'-strand, and no longer cuts DNA on the 3'-strand downstream of Chi, generating a long 3'-terminated ssDNA overhang (*Dixon and Kowalczykowski, 1993*; *Anderson and Kowalczykowski, 1997a*). Moreover, Chi recognition imparts RecBCD with the ability to load RecA protein onto the 3'-tailed product to produce a nucleoprotein filament that engages in strand exchange with the homology donor (*Anderson and Kowalczykowski, 1997b*). Chi sequences are highly over-represented in host genomes but are often lacking in bacteriophage DNA (*Dillingham and Kowalczykowski, 2008*; *El Karoui et al., 1999*; *Tracy et al., 1997*; *Cardon et al., 1993*). Therefore, regulation of RecBCD by Chi has been viewed as a form of self-recognition which instructs RecBCD to repair host DNA but to destroy and catalogue foreign DNA (see also *Subramaniam and Smith, 2022* for an opposing view).

Bacteriophage could evade the immunity function of RecBCD in several different ways (*Bobay et al., 2013*). One strategy is to select for Chi sequences within their genome, but this is thought to incur a significant fitness penalty: phage are themselves dependent on recombination for packaging and genetic variation, and the use of the RecBCD pathway may provide suboptimal outcomes and limit the breadth of possible hosts. Instead, many phage express inhibitors of RecBCD which enable them to avoid genome degradation and prioritise the use of their own autonomous recombination system; the phage-encoded exonuclease (*exo*) and recombinase (*bet*) genes. A well-characterised example of this strategy is provided by coliphage $\lambda$ Gam protein. Gam acts as a DNA mimic protein to bind directly to RecBCD and inhibit all of its activities by competing for the DNA binding site (*Court et al., 2007*; *Murphy, 1991*; *Murphy, 2007*; *Poteete et al., 1988*; *Wilkinson et al., 2016b*). Analogous RecBCD inhibitors are also present in phage T4, T7, and SPP1, but these do not share sequence identity with Gam and are less well-characterised in comparison (*Pacumbaba and Center, 1975*). The T7 gp5.9 protein, which is one subject of this work, was discovered as a protein responsible for reduced ATP-dependent exonuclease activity in extracts from cells infected with T7 and was subsequently shown to reduce the activity of purified RecBCD in vitro (*Pacumbaba and Center, 1975*; *Lin, 1992*). Despite the lack of any detectable homology to $\lambda$ Gam, gp5.9 is also a small and highly acidic protein, both of which are properties associated with DNA mimicry (*Wang et al., 2019*; *Wang et al., 2014*). Therefore, T7 gp5.9 may act via a similar mechanism to Gam by competing for the RecBCD DNA binding locus. Remarkably, bacterial strains containing the Ec48 retron system have turned this RecBCD evasion strategy to their own advantage (*Millman et al., 2020*). Within such cells, RecBCD interaction with Gam or gp5.9 acts as a sensor of phage infection and triggers cell suicide, protecting the whole bacterial population at the expense of infected cells.

A completely different strategy to evade the anti-phage activities of RecBCD is exemplified by the Abc2 protein of *Salmonella* phage P22 (*Poteete et al., 1988*; *Murphy, 1994*; *Murphy, 2000*; *Murphy, 2012*; *Murphy and Lewis, 1993*). Abc2 binds to the RecC subunit, from where it is proposed to hijack RecBCD to promote recombination (rather than degradation) of phage. At the molecular level, this might be achieved by promoting the conversion of RecBCD into its recombinogenic mode and adapting the RecA-loading function for the phage-recombinase. This possibility is supported by several observations: phage encoding Abc2-like proteins apparently lack an exonuclease function (which would instead be provided by the co-opted RecBCD complex), Abc2 expression eliminates the SOS response to DSBs which requires RecA loading, and Abc2-RecBCD fails to respond to Chi yet displays properties somewhat similar to the Chi-modified enzyme constitutively (i.e. upregulated 5'-nuclease activity).

In this work, we present a comparative study of the interactions between RecBCD and the phage-encoded gp5.9 and Abc2 proteins. We find that there are stark differences in the effects of the two proteins on RecBCD activity. Whereas gp5.9 inhibits all tested activities of RecBCD, Abc2 has no significant effect on the ability of the complex to bind and unwind DNA. Accordingly, cryoEM structures of

gp5.9-RecBCD and Abc2-RecBCD show that the two phage proteins bind to entirely different parts of the RecBCD structure. The gp5.9 protein uses DNA mimicry to bind the RecB arm domain and compete for the interaction between RecBCD and DNA ends. The Abc2 protein, together with a host encoded prolyl-isomerase, binds to the RecC subunit from where it might influence the recombinase loading step of homologous recombination.

## Results

### Gp5.9, but not Abc2, inhibits RecBCD helicase activity

The T7 gp5.9 protein was shown previously to inhibit RecBCD, but the mechanism was not determined. Therefore, we made recombinant gp5.9 and studied its effects on RecBCD activity in vitro. We found that gp5.9 expression was toxic to *E. coli*, and therefore used insect cell expression to express and purify the protein with a cleavable histidine tag (*Figure 1—figure supplement 1*). Unless stated otherwise, all experiments were performed with a preparation of gp5.9 with the tag removed. We first assessed the effect of T7 gp5.9 on RecBCD helicase-nuclease activity using a simple gel-based DSB resection assay. As expected, RecBCD rapidly unwound a linearised duplex DNA substrate (*Figure 1a*, lanes 1–4). In the presence of excess purified gp5.9 (1 µM), the RecBCD enzyme was strongly inhibited (compare lanes 4 and 5). To assess the specificity of gp5.9 for RecBCD, we performed analogous experiments using the AddAB helicase-nuclease, a RecBCD orthologue from *Bacillus subtilis* (*Yeeles and Dillingham, 2010*), and observed no inhibition of DNA degradation (lanes 6–10). This strongly suggests that gp5.9 acts by binding directly and specifically to the RecBCD complex, as opposed to blocking DNA ends.

We next compared the effects of Abc2 on RecBCD activity. It has been established previously that Abc2 binds directly to RecBCD but, in comparison to gp5.9 and Gam, it seems to have only moderate effects on its DNA binding, helicase, and nuclease activities in vitro (*Murphy, 2000*). We were unable to purify the Abc2 polypeptide alone. Therefore, to confirm these observations, we purified the intact RecBCD-Abc2 complex which was found to co-purify with host peptidyl-prolyl *cis-trans* isomerase B (*E. coli* PpiB) as observed previously (*Court et al., 2007*; *Murphy, 1991*; *Figure 1—figure supplement 1*). It was found that the expression of the Pro68Ala mutant form of Abc2 (Abc2[P68A]) with RecBCD prevented co-purification with the host PpiB protein, allowing us to also isolate the RecBCD-Abc2[P68A] complex (*Figure 1—figure supplement 1*). Therefore, in the following experiments, we compare the biochemical activities of independently purified samples of free RecBCD, RecBCD-Abc2[P68A], and the RecBCD-Abc2-PPI complex. As expected based on previous work, the observed DNA unwinding activities of the three preparations were very similar (*Figure 1a*, compare lanes 1–4 with 11–14 and 16–19), and all three preparations were completely inhibited by excess gp5.9 (lanes 5, 15, and 20).

To provide more quantitative insights into RecBCD inhibition by gp5.9, we next employed a continuous multiple turnover helicase assay which uses a fluorescent biosensor for ssDNA (fSSB) to monitor DNA unwinding in real time (*Chisty et al., 2018*; *Dillingham et al., 2008*; *Hedgethorne and Webb, 2012*; *Figure 1b*). RecBCD was pre-incubated at a low concentration (10 pM) with a linear DNA substrate (100 pM molecules) that was devoid of Chi sequences. DNA unwinding was then initiated by the addition of ATP in the presence of fSSB. A rapid increase in fluorescence indicated that RecBCD unwound the DNA over several hundred seconds (*Figure 1c*). We next repeated the experiment in the presence of increasing concentrations of gp5.9, measuring the initial rate of unwinding in each case. Increasing concentrations of gp5.9 inhibited RecBCD helicase and a plot of [gp5.9] versus unwinding rate was well-fit to an inhibitor dose–response curve yielding $IC_{50} = 1.3$ nM (solid line; *Figure 1d*). Equivalent experiments using a gp5.9 preparation retaining the histidine tag yielded a significantly higher $IC_{50}$ value of 23 nM (dotted line; *Figure 1d*). This suggests that the N-terminal region of the protein is important functionally. Together, these experiments show that both gp5.9 and Abc2 bind directly to RecBCD, but only gp5.9 acts as an inhibitor of RecBCD DNA unwinding activity.

### Gp5.9 inhibits RecBCD-DNA interaction independently of Abc2 binding

To determine the mechanistic basis for inhibition of RecBCD by gp5.9, and to assess whether the phage proteins could bind simultaneously to RecBCD, we next performed EMSA and 'inverse EMSA' experiments to monitor DNA binding by different RecBCD-inhibitor complexes. In conventional EMSA experiments, a Cy5-labelled DNA substrate (25mer blunt duplex) at low concentration (5 nM)

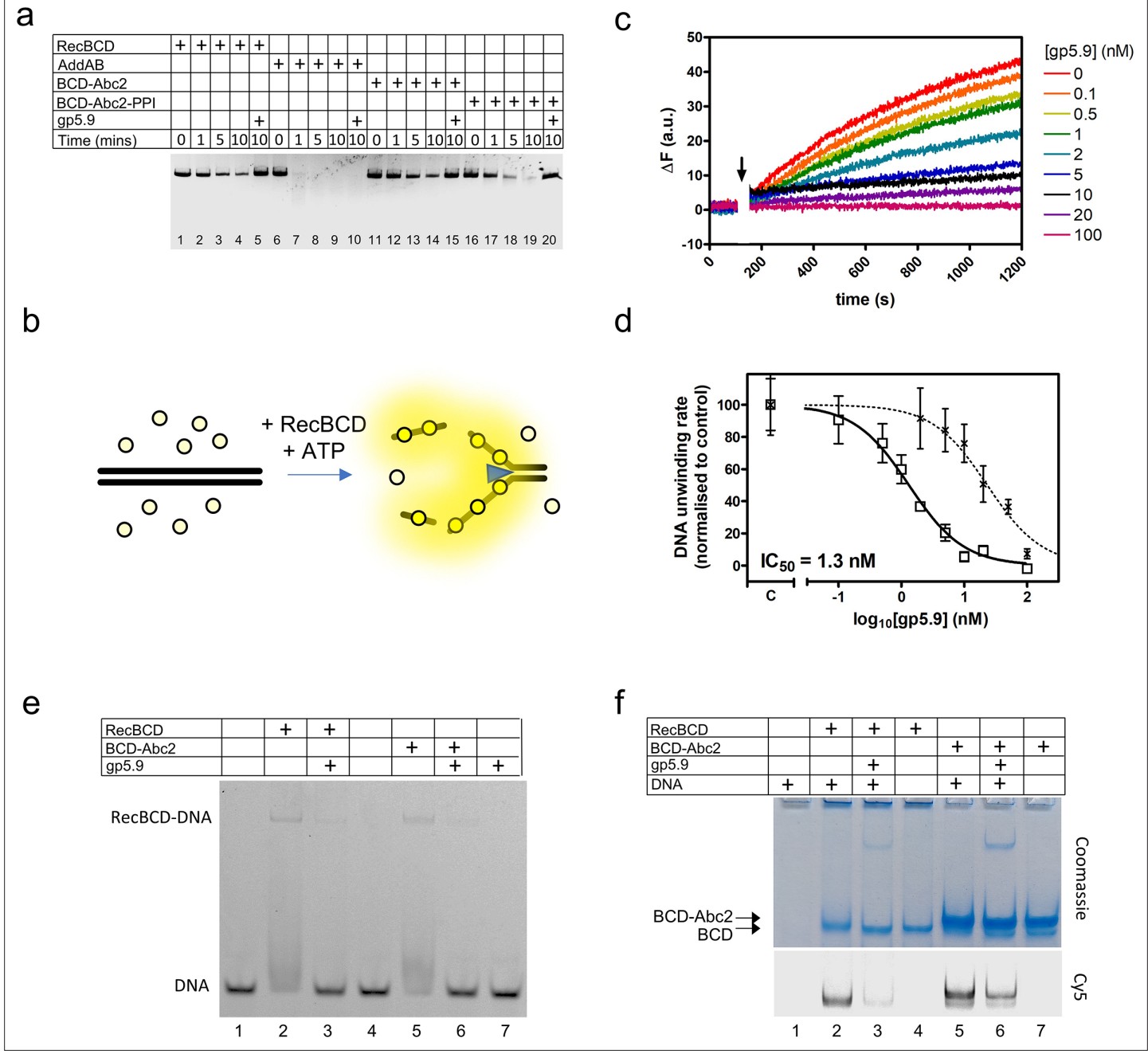

**Figure 1.** RecBCD helicase activity is inhibited by gp5.9 but not Abc2. (**a**) Gel-based double-stranded DNA break (DSB) resection assay. The indicated proteins or protein complexes were incubated with a linearised dsDNA substrate for the indicated times in the presence of ATP. RecBCD (or AddAB) activity leads to DNA strand separation and depletion of the substrate. Substrate depletion is not affected by the presence of Abc2 or PPI. The substrate is not unwound by RecBCD in the presence of gp5.9 regardless of the presence of Abc2/PPI, but gp5.9 does not inhibit the orthologous AddAB complex. (**b**) Schematic of the DNA helicase assay using the fSSB biosensor (yellow circles). The ssDNA products of DNA unwinding are bound by fSSB causing a fluorescence increase. (**c**) Representative example of DNA unwinding traces for RecBCD in the presence of increasing [gp5.9]. The black arrow indicates addition of ATP to initiate the reaction. (**d**) A plot of initial DNA unwinding rate versus [gp5.9] yields a low nanomolar inhibition constant ($IC_{50}$). The two data sets shown are for gp5.9 with the N-terminal his-tag removed (squares and solid fit line; $IC_{50}$ = 1.3 nM) and with the his-tag intact (crosses and dotted fit line; $IC_{50}$ = 23 nM). Values for the unwinding rate (mean and standard error for three technical repeats) are normalised to a control experiment in the absence of gp5.9 (marked C on the x-axis). (**e**) Electrophoretic mobility shift assay showing the effect of gp5.9 and Abc2 on the DNA binding activity of RecBCD. A Cy5-labelled blunt-ended duplex DNA substate was incubated with the proteins indicated and then run on a native gel to separate the free and bound DNA species. (**f**) Native 'inverse EMSA' assay showing binding of DNA by RecBCD or RecBCD-Abc2 complexes, and the effect of gp5.9 on both systems. The indicated complexes were incubated with or without labelled DNA substrate either in the presence or absence

*Figure 1 continued on next page*

*Figure 1 continued*

of gp5.9 as indicated. The samples were then run on native polyacrylamide gels and stained with Coomassie to reveal the position of the intact protein complexes. The same gel was also imaged for Cy5 fluorescence to reveal the position of the DNA. Note that free DNA runs off the bottom of the gel in this experiment, whereas bound DNA co-migrates with the RecBCD complexes. All gel images are representative examples of experiments that were technically reproduced at least once.

The online version of this article includes the following source data and figure supplement(s) for figure 1:

**Source data 1.** Compresssed file containing images of uncropped gels and raw data associated with *Figure 1*.

**Figure supplement 1.** Purified proteins and protein complexes used in this study.

was incubated with RecBCD or RecBCD-Abc2 either in the presence or absence of excess gp5.9 (1 μM). RecBCD caused a shift in the mobility of the substrate indicative of the expected RecBCD-DNA interaction (*Figure 1e*). This gel shift was largely eliminated by the addition of gp5.9 to RecBCD before DNA was added. The RecBCD-Abc2 complex behaved identically to wild type. It was able to bind the duplex DNA substrate, but binding was largely inhibited by excess gp5.9. The phage protein alone did not interact with DNA. In complementary 'inverse' EMSA experiments (*Figure 1f*), RecBCD-DNA or RecBCD-Abc2-DNA complexes were run at high concentrations in native polyacrylamide gels that were imaged for Cy5-DNA using a confocal scanner and then stained with Coomassie to detect protein-containing complexes. Note in these experiments that free DNA is not detected as it has run off the bottom of the gel (lane 1). RecBCD was able to bind to DNA as expected (lane 2). However, addition of gp5.9 before DNA dramatically reduced the amount of DNA detected and caused a small increase in the mobility of the RecBCD complex showing it was interacting directly (compare lanes 2 and 3). The RecBCD-Abc2 complex displayed a reduced mobility compared to RecBCD alone (compare lanes 2–4 with 5–7). There was evidence for a small amount of free RecBCD in the preparation in the form of a fine band running with identical mobility to the RecBCD-alone preparation. Importantly, both bands co-migrated with Cy5-DNA showing that DNA and Abc2 can bind simultaneously to RecBCD (lane 5). Pre-addition of gp5.9 to the RecBCD-Abc2 preparation reduced Cy5-DNA binding to both protein complexes (compare lanes 5 and 6). Together, these data show that the gp5.9 binding site on RecBCD is distinct from that of Abc2 and that gp5.9 inhibits DNA binding, potentially by direct competition at the DNA binding locus.

## gp5.9 is a DNA mimic protein with an unprecedented architecture

We next purified the RecBCD-gp5.9 complex (using the tag-free form of gp5.9) and analysed its structure using single-particle cryoEM. The dataset was found to be relatively homogeneous with evident additional density for the phage protein on the surface of the complex (*Figure 2—figure supplement 1*). There was structural heterogeneity only within the RecD protein, suggesting that the RecD conformation becomes uncoupled to the rest of the complex when bound to gp5.9. This was improved by 3D classification, after which an ordered high-resolution class (30% of the classified particles) was separated for the RecBCD-gp5.9 complex with the RecD 1A and 1B domains resolved but not the 2A nor SH3 domains (*Figure 2—figure supplement 1e*). A 3.2 Å resolution cryoEM map was obtained (*Figure 2a*, *Figure 2—figure supplement 1h*) facilitating the building of a model for two chains of gp5.9, both containing residues 1–49 (*Figure 2b*, *Figure 2—figure supplement 2*). The structure reveals that gp5.9 adopts a parallel coiled-coil architecture in which the N-termini are slightly braced apart by a short anti-parallel beta-sheet. Many aspartate and glutamate residues are displayed on the outer surface of this exceptionally negatively charged protein (pI = 4.0; *Figure 2b and c*). The phage protein engages RecBCD in a position which overlaps extensively with the DNA binding site and also with the binding site for the DNA mimic protein Gam (*Figure 2d–f*). The C-terminal end of the coiled-coil binds to the extended RecB arm domain, while the N-terminus binds closer to the RecBCD core, making contacts with both the RecB and RecC subunits and helping to explain the reduced efficacy of inhibition by purified gp5.9 retaining an N-terminal tag (*Figure 1d*). The overall structures of RecBCD when bound to either DNA or gp5.9 are almost identical, although small rigid body domain movements help accommodate differences in the dimensions of the two ligands (*Figure 2—figure supplement 3*).

Many ion pair contacts are formed between D/E and K/R residues in gp5.9 and RecBCD, respectively (*Figure 3*, *Table 1*). These imitate a subset of the contacts formed between the negatively

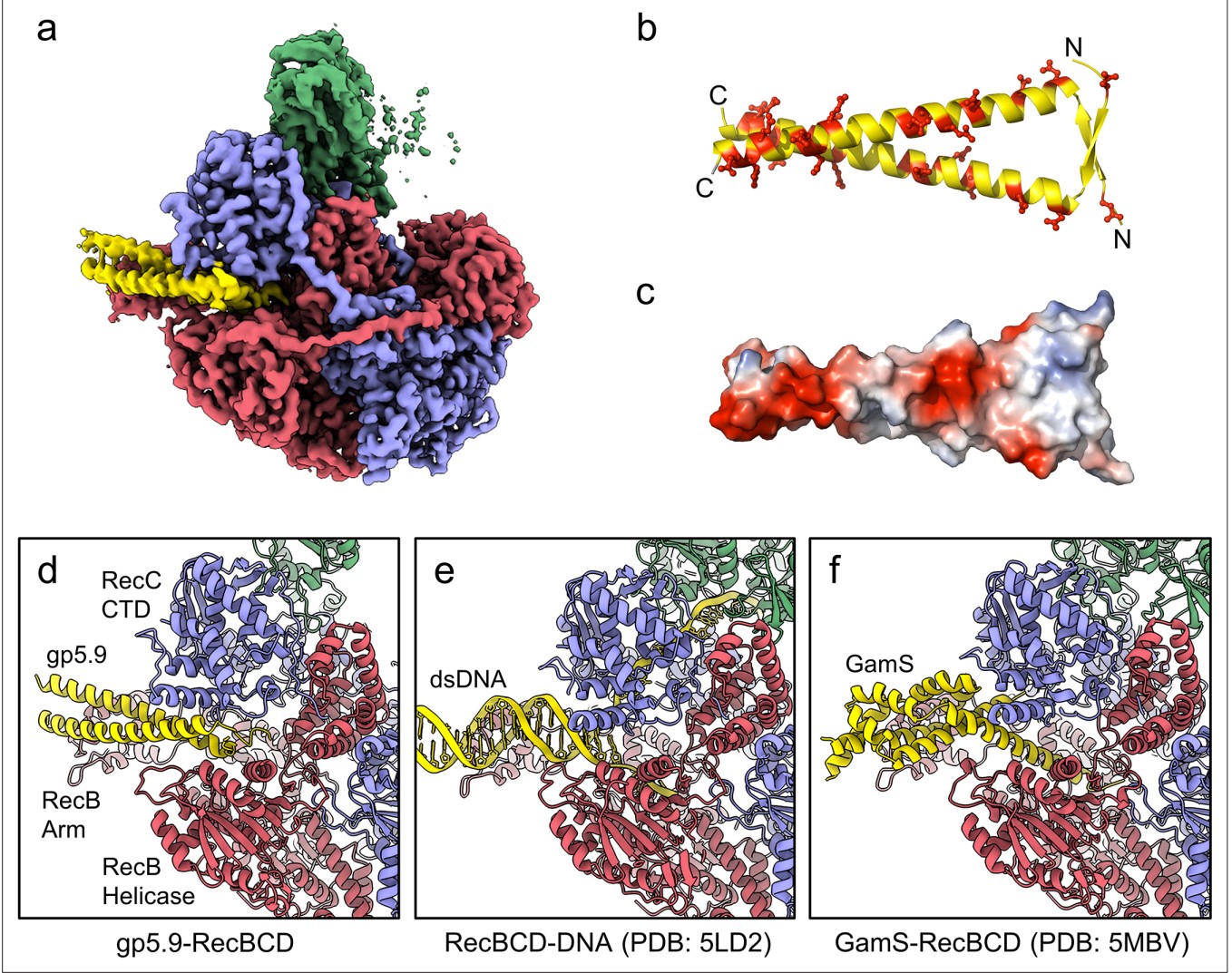

**Figure 2.** CryoEM structure of the RecBCD-gp5.9 complex. (**a**) CryoEM map of the RecBCD-gp5.9 complex. Subunit colour coding is as follows: RecB in red, RecC in slate blue, RecD in green, and gp5.9 in yellow. (**b, c**) The gp5.9 dimer adopts a parallel coiled-coil architecture braced by an N-terminal beta-sheet and displays many negatively charged Asp and Glu residues (red) on its surface. (**d**) Section of the atomic model of the RecBCD-gp5.9 complex with colouring as in (**a**). (**e, f**) Sections of the models of the RecBCD-DNA (PDB:5LD2) and RecBCD-Gam (PDB:5MBV) complexes for comparison.

The online version of this article includes the following figure supplement(s) for figure 2:

**Figure supplement 1.** CryoEM processing scheme for the RecBCD-gp5.9 dataset.

**Figure supplement 2.** Modelling the structure of the gp5.9 phage protein.

**Figure supplement 3.** Small rigid body domain movements in RecBCD facilitate binding to either DNA or gp5.9.

**Figure supplement 4.** Rebuilding of the RecB arm domain model based on the high-resolution RecBCD-gp5.9 cryoEM map.

charged DNA phosphates and K/R residues in the RecBCD-DNA complex and, on this basis, we can conclude that gp5.9 acts by DNA mimicry. Interestingly, the contacts formed between RecBCD and Gam (another DNA mimic protein) are equivalent to a different subset of the RecBCD-DNA interactions such that only a few of the charge-based interactions (with RecB Arg residues 254, 255, and 761) are conserved across all three complexes (see highlighted residues in *Figure 3* and *Table 1*). Indeed, besides their dimeric, α-helical, and highly charged structures there is little else in common between the two phage proteins and the coiled-coil architecture of gp5.9 is unprecedented for a DNA mimic. In addition to charge-based interactions that mimic contacts with DNA phosphates, gp5.9 makes

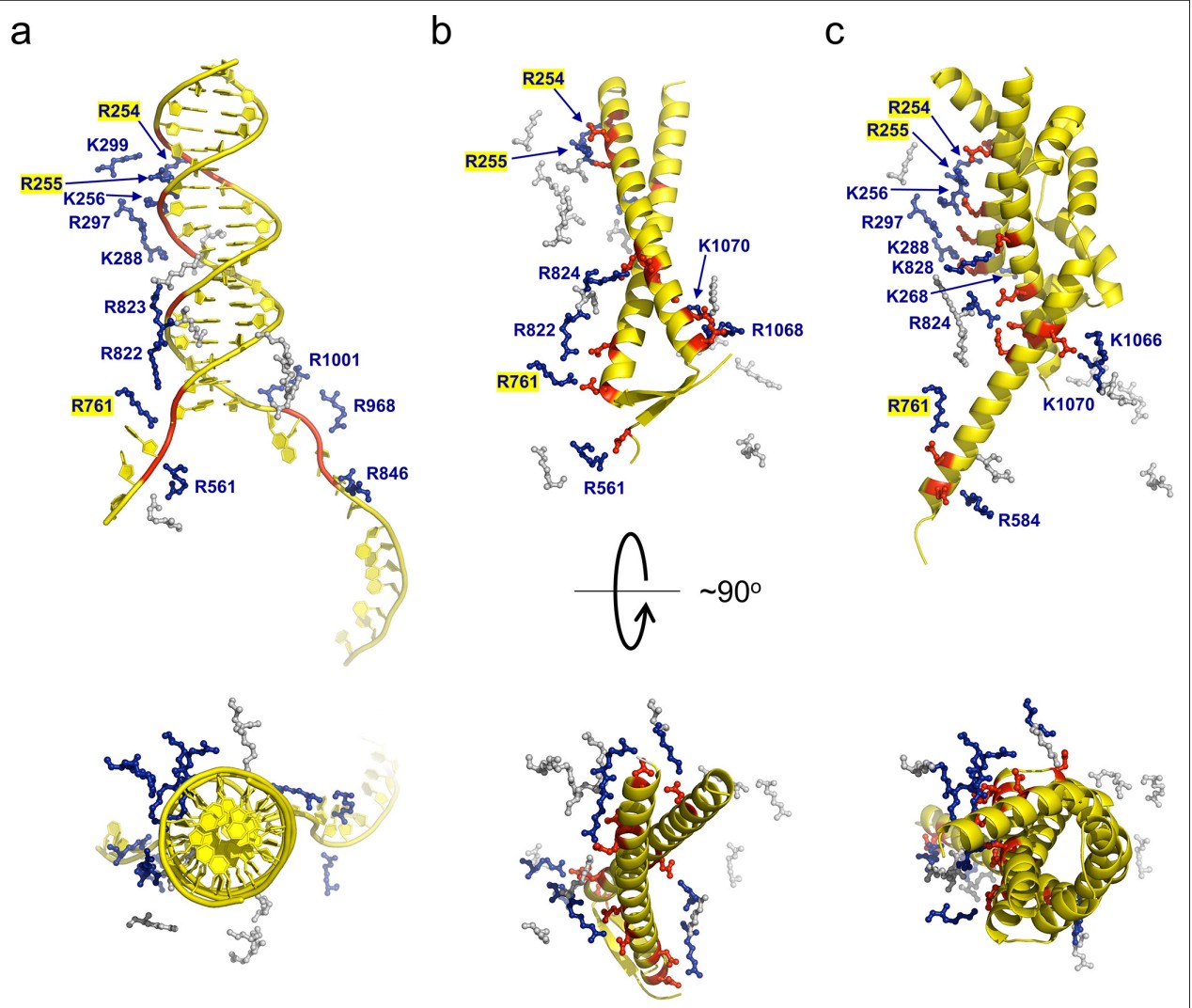

**Figure 3.** gp5.9 is a DNA mimic protein. Charge pair interactions between R/K residues in RecBCD (blue) and either phosphates or D/E residues (red) for DNA (**a**), gp5.9 (**b**), or Gam (**c**). The images are all taken from the same point of view for complexes that were superimposed using the RecB arm domain. The R/K residues displayed are all of those shown in *Table 1* and are the same for all three complexes. They are coloured blue and labelled with residue numbers if they make contacts with the ligand in each case (either DNA, gp5.9, or Gam) and are grey and unlabelled if they do not.

The online version of this article includes the following source data for figure 3:

**Source data 1.** Complete list of interactions between the RecBCD complex and gp5.9.

additional protein–protein interactions which presumably provide specificity for RecBCD (*Figure 3—source data 1*), as opposed to other DNA binding proteins including AddAB.

## Abc2 binds to the RecC subunit, close to a putative exit channel for recombinogenic DNA

We next solved cryoEM structures of the RecBCD-Abc2-PPI complex bound to a tailed DNA substrate (*Figure 4*). All of the particles within the dataset contained ordered density for RecBCD with additional density evident for the Abc2 protein (*Figure 4—figure supplement 1*). Focused classification with a mask around the phage binding site (*Figure 4—figure supplement 1f*) isolated 71% of the particles into a homogeneous class from which a 3.4 Å resolution structure of the RecBCD-Abc2-DNA complex was solved (*Figure 4a*, *Figure 4—figure supplement 1g*). A second class containing 22% of the data displayed further additional resolved density in the vicinity of Abc2, corresponding to the *E. coli* PpiB protein. From this class, a 3.8 Å resolution structure of RecBCD-Abc2-PPI-DNA complex was

**Table 1.** Gp5.9 is a DNA mimic protein.
Ion pair contacts (<4 Å) between Arg/Lys residues in the RecBCD complex and negatively charged moieties in either DNA (phosphates) or DNA mimic proteins (Asp/Glu). Shaded rows highlight charge–charge interactions that are analogous across all three complexes. These interactions are illustrated in *Figure 3*.

| RecBCD (subunit indicated) | DNA | Gp5.9 | Gam |
|---|---|---|---|
| R254 (B) | O3'-25 | E45 | D73 |
| R255 (B) | O3'-56 | D38/E39 | E118/E125 |
| K256 (B) | OP1-57/O5'-57 | | E118 |
| K264 (B) | | E36 | |
| K268 (B) | | | D107 |
| K288 (B) | OP-59 | | E111 |
| R297 (B) | OP-58 | | E114 |
| K299 (B) | OP-27 | | |
| R561 (B) | OP-69 | D4 | |
| R584 (B) | | | D48 |
| R761 (B) | O5'–68/O3'-67 | D11 | E51 |
| R822 (B) | OP-67 | D15 | |
| R823 (B) | OP-18 | | |
| R824 (B) | | D21/E24 | E65/E68/E125 |
| K828 (B) | | | E118 |
| R846 (C) | OP-10/O3'-9 | | |
| R968 (C) | OP-11/O3'-10 | | |
| R1001 (C) | OP-12 | | |
| K1066 (C) | | | E70/D74 |
| R1068 (C) | | D11/D15 | |
| K1070 (C) | | E24 | E70 |

solved (*Figure 4b*, *Figure 4—figure supplement 1h*). The Abc2 protein adopts an extended alpha-helical structure and binds to the surface of the 'inactivated helicase domains' of RecC which play a key role in Chi recognition (*Figure 4c–e*, *Figure 4—source data 1*; *Murphy, 2000*). Consistent with biochemical observations, this location suggests a role for Abc2 in modulating the late stages of the DSB processing reaction catalysed by RecBCD (i.e. Chi recognition and associated conformational changes mediated by a 'latch' structure in RecC, or the subsequent loading of RecA protein). The binding site is also close to the interface with the helicase domains of RecB, and to a tunnel between RecB and RecC (*Figure 4—figure supplement 2*) which has been suggested to allow exit of a recombinogenic ssDNA loop (the site of RecA loading) after Chi recognition (*Cheng et al., 2020*; *Yang et al., 2012*). In the RecBCD-Abc2-PPI structure, RecBCD and Abc2 are essentially identical, but Abc2 is also bound to the host PpiB protein. Abc2 interacts extensively with the Chi-recognition domains of RecC as before, but there are no interactions between PpiB and RecBCD (*Figure 4c*). The structure can be superposed onto the corresponding translocation activated RecBCD-DNA structure (PDB: 5LD2; *Wang et al., 2019*), containing the same DNA substrate and non-hydrolysable ATP analogue, ADPNP (*Figure 4—figure supplement 3a*). This shows that Abc2 has no significant effect on the conformation of the majority of the RecBCD complex except for a minor opening of the nuclease domain relative to RecC and RecD. Additionally, Abc2 binding displaces a helical bundle, residues 252–294, from the RecC surface (*Figure 4—figure supplement 3b*) and some partial density can be seen for the displaced region clamping back over the exposed face of Abc2 (*Figure 4—figure supplement 3c*).

The C-terminal region of Abc2 is disordered to differing extents in the structures; helix-2 of Abc2 is ordered up to residue 52 on the surface of RecC (*Figure 4—figure supplement 4a*) but in the PpiB-bound state this helix is extended with Abc2 modelled up to residue 66 (*Figure 4—figure supplement 4b*). This C-terminal extension protrudes into the PpiB density, into which a high-resolution crystal structure of PpiB containing a bound substrate analogue (PDB: 1LOP; *Konno et al., 1996*) can be docked (*Figure 4f*). The resulting model strongly suggests that P68 from Abc2 would occupy the *cis-trans* isomerisation active site in PpiB. A *cis* proline conformation at this position would disrupt the helical secondary structure and potentially direct the disordered C-terminus of Abc2 towards the RecB-RecC interface. In fact, when the RecBCD-Abc2-PPI map is blurred and contoured down, a continuous weak density path can be observed for the continuation of the Abc2 C-terminus through the PpiB binding cleft and out towards the RecB helicase domains (*Figure 4—figure supplement 4c*). Looking around the RecB helicase domains, additional unaccountable patches of density are found above the noise in-between the 1A

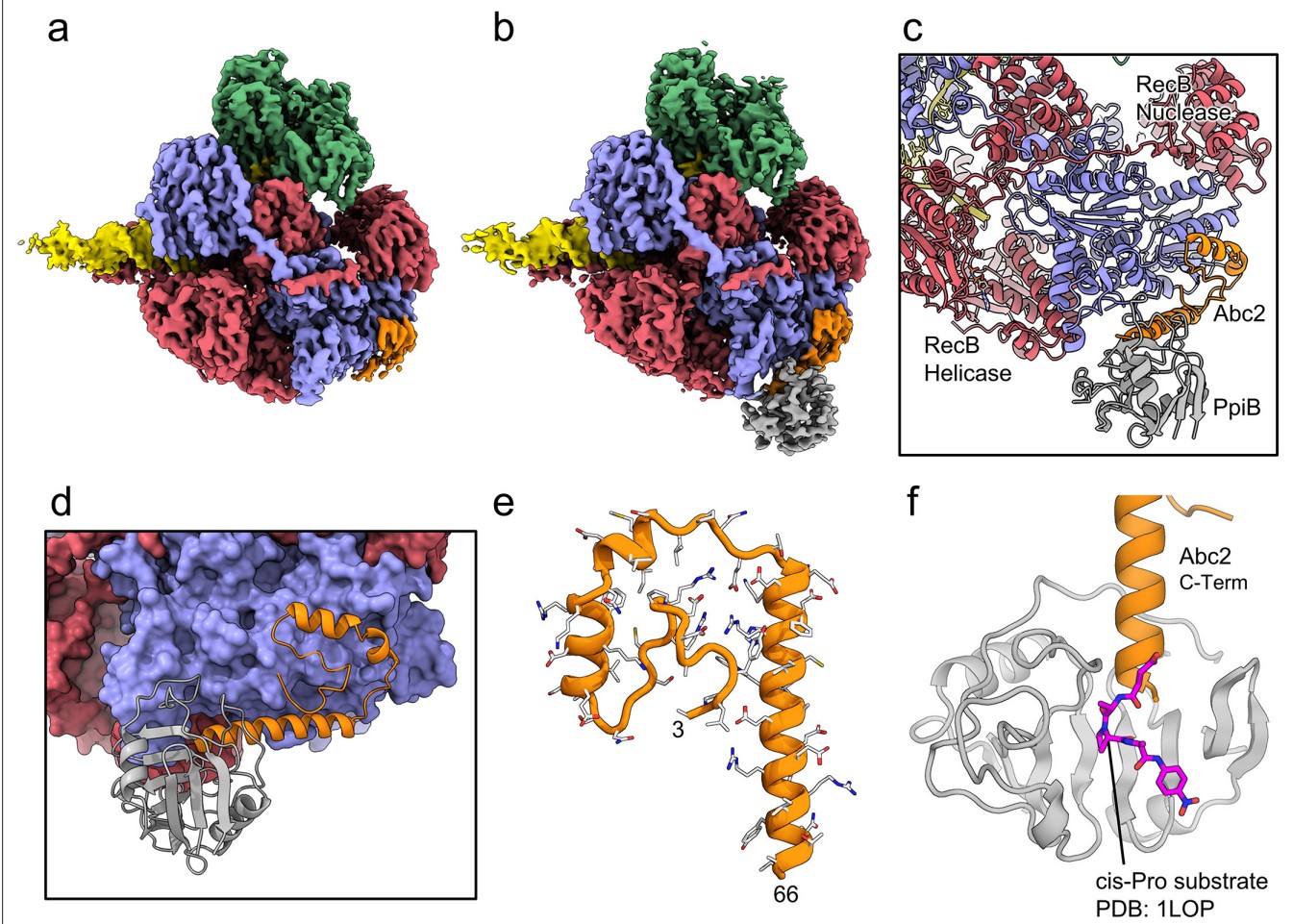

**Figure 4.** CryoEM structures of the RecBCD-Abc2 and RecBCD-Abc2-PPI complexes. (**a**) CryoEM map of the RecBCD-Abc2-DNA complex, coloured as in *Figure 2* but with the DNA substrate yellow and Abc2 in orange (**b**) CryoEM map of the RecBCD-Abc2-PpiB-DNA complex, with PpiB in grey. (**c**) Section of the atomic model of the RecBCD-Abc2-PpiB-DNA complex. (**d**) Surface view of the Abc2 binding site on the surface of the RecC Chi-binding domains. (**e**) Atomic model of residues 3–66 of Abc2 in the PpiB-bound complex showing an extended helical structure. (**f**) Model of the C-terminus of Abc2 bound to PpiB. A substrate analogue (magenta) is superimposed on the PpiB structure based upon PDB: 1LOP. The structure strongly suggests that P68 (two residues beyond the final modelled amino acid 66) would reside in the *cis-trans* isomerisation active site.

The online version of this article includes the following source data and figure supplement(s) for figure 4:

**Source data 1.** Complete list of interactions between the RecBCD complex and Abc2.

**Figure supplement 1.** CryoEM processing scheme for the RecB^DCD-Abc2-PPI-DNA dataset.

**Figure supplement 2.** Overview of the location of Abc2 binding in the context of the RecBCD complex.

**Figure supplement 3.** Abc2 binding induces minimal conformational changes in the RecBCD complex.

**Figure supplement 4.** Modelling of Abc2 and PpiB in the cryoEM maps.

**Figure supplement 5.** Additional unmodelled density suggests a path for the Abc2 C-terminus breaching the RecB helicase domains.

**Figure supplement 6.** AlphaFold modelling of the full Abc2 protein shows a divergent C-terminus.

and 2A motor domains which enclose the ATP-binding site (*Figure 4—figure supplement 5a*). This additional density is not seen in the Abc2 complex without bound PpiB (*Figure 4—figure supplement 5b*). AlphaFold (*Yüksel et al., 2016*) models of the Abc2 C-terminus are highly divergent and cannot be used to interpret the density (*Figure 4—figure supplement 6*).

## Discussion

In this work, we used biochemical and structural methods to study the interactions of two phage proteins, T7 gp5.9 and P22 Abc2, with the *E. coli* RecBCD complex. We found that low nanomolar concentrations of gp5.9 specifically inhibit RecBCD helicase activity in vitro by preventing DNA substrate binding. Our structure of the RecBCD-gp5.9 complex explains the biochemical observations by showing how gp5.9 competes directly for the DNA binding site through mimicry of the natural DNA substrate.

The T7 gp5.9 protein is functionally analogous to Gam from phage $\lambda$ (*Murphy, 2007*). Moreover, both are small, dimeric, and highly acidic proteins which form a compact and predominantly alpha-helical structure (*Court et al., 2007*). Despite these similarities, their primary structures, overall folds, and the molecular-level details of their interaction with RecBCD are different. Indeed, DNA mimic proteins display a remarkable structural diversity (*Wang et al., 2019*; *Wang et al., 2014*) and, to the best of our knowledge, the coiled-coil architecture observed here for gp5.9 has not been observed in nature before this study. Interestingly, however, this fold has been used previously as the structural framework for the rational design of synthetic DNA mimics targeting restriction endonucleases (*Yüksel et al., 2016*). Our work here validates the choice of coiled-coils for producing such synthetic proteins and may help inform their design against novel targets.

In cells containing the bacterial retron Ec48, the presence of either gp5.9 or Gam (either of which are indicative of phage infection) triggers programmed cell death as an abortive infection mechanism (*Millman et al., 2020*). Because the Ec48 system is not tolerated in a Δ*recB* mutant, and because Gam was known to bind mainly to RecB, it was suggested that the retron binds directly to RecB and its release is triggered by Gam or gp5.9 to activate downstream effectors of cell suicide by unknown mechanisms. This model implies that Gam and gp5.9 bind to the same interface on RecB. Our results here are consistent with this hypothesis because Gam and gp5.9 do indeed share the same binding location on the RecB arm domain and other nearby regions of RecB and RecC. Therefore, DNA, Gam, gp 5.9, and possibly the retron all compete for the same site on RecBCD. In retron-free *E. coli* cells, expression of gp5.9 likely sequesters all of the available RecBCD complex present at only ~10 copies per cell (*Lepore et al., 2019*; *Taniguchi et al., 2010*; *Taylor and Smith, 1980*) in an inactive form, thereby protecting T7 DNA from degradation and prioritising the use of phage-recombinases (*Figure 5*). In addition to its natural roles, gp5.9 may also find use as a biotechnology tool for enhancing recombineering efficiency, improving cell-free transcription-translation systems, or stabilising DNA nanostructures in vivo through its ability to potently inhibit RecBCD exonuclease activity (*Sitaraman et al., 2004*; *Klocke et al., 2018*; *Yu et al., 2000*).

The effects of Abc2 on RecBCD activity are complex and remain less well understood than is the case for the two DNA mimic proteins. Expression of Abc2 in *E. coli* eliminates RecBCD-dependent recombination but exonuclease activity is retained (*Murphy and Lewis, 1993*). We found here that there is no substantial effect of Abc2 on RecBCD helicase activity in vitro. This stands in stark contrast to the complete inhibition of all activities afforded by interaction with gp5.9. Our results differ somewhat from previous biochemical studies (*Poteete et al., 1988*; *Murphy, 1994*; *Murphy, 2000*; *Murphy, 2012*; *Murphy and Lewis, 1993*) which showed that Abc2 had modest effects on RecBCD DNA binding (two- to fourfold tighter) and helicase-nuclease activity (three- to fourfold lower) in vitro. Such differences are potentially explained by known prep-to-prep variability in RecBCD-specific activity. Importantly, however, it was also shown that the enzyme failed to respond to Chi and displayed properties that are somewhat akin to a constitutively Chi-activated complex (upregulated 5′-exonuclease activity). Based upon their observations, Murphy and co-workers developed a model in which Abc2 hijacks RecBCD to generate the ssDNA required for phage recombination and modifies its RecA loading function to instead recruit its own recombinase (called Erf) to the ssDNA products (*Figure 5*). Our biochemical and structural data are consistent with this hypothesis. Abc2 is bound to RecC in close proximity to both a 'latch helix' which is thought to control conformational changes in response to Chi, and to an 'alternative exit channel' in the RecBCD complex which is thought to accommodate a ssDNA loop in the post-Chi state (i.e. the recombinogenic mode) (*Cheng et al., 2020*; *Yang et al., 2012*; *Handa et al., 2012*). Under normal circumstances, this loop would be loaded with *E. coli* RecA protein to promote subsequent steps in the host DNA break repair pathway. Given its location, it is plausible that Abc2 prevents RecA loading at this site and/or promotes the binding of Erf to the emerging ssDNA, but this remains to be tested experimentally. In addition to Abc2 and Erf, the P22

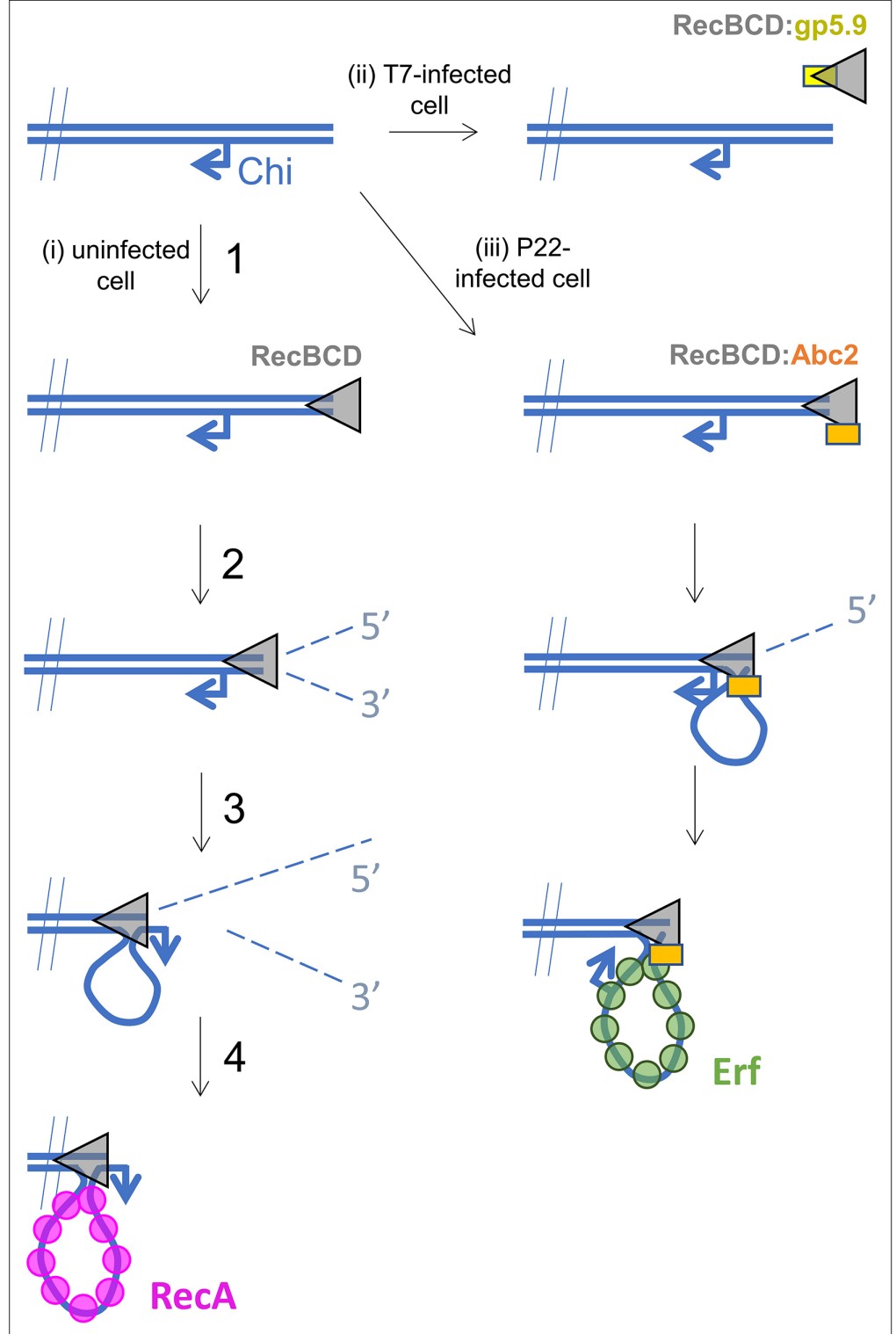

**Figure 5.** Hypothetical models for control of RecBCD by T7 gp5.9 and P22 Abc2. In uninfected cells (pathway i), free DNA ends are processed by RecBCD to yield a recombinogenic 3'-terminated ssDNA overhang coated with host RecA protein. The reaction proceeds in five steps. (1) RecBCD binds to the free DNA end. (2) RecBCD translocates and unwinds the duplex DNA degrading both strands as it progresses into fragments of single-stranded DNA of varying lengths. This activity is detrimental to phage as it can help degrade invading linear DNA species and initiates CRISPR acquisition. (3) RecBCD recognises a Chi sequence (these are over-represented in host DNA). (4) RecBCD continues to unwind DNA and to degrade the 5'-terminated DNA strand, but the

*Figure 5 continued on next page*

*Figure 5 continued*

3'-terminated strand is protected from degradation and forms an expanding loop structure. (5) RecBCD loads RecA protein onto the ssDNA loop ready for subsequent strand invasion and further steps of DNA repair. In T7-infected cells (pathway ii), the gp5.9 protein titrates RecBCD by competing for the DNA binding site. Linear phage DNA cannot be degraded by RecBCD and is available for phage recombination and replication. In P22-infected cells (pathway iii), Abc2 binds to the RecBCD complex and modifies the enzyme to load the P22 recombinase (Erf), as opposed to RecA, onto the ssDNA products. Recombinase loading might occur either constitutively as indicated here or in response to Chi-recognition as in the unperturbed pathway.

recombination system also includes the uncharacterised proteins Abc1 (Anti-RecBCD protein 1) and Arf (Accessory recombination factor) which are both required for full activity during P22 infection (*Murphy, 2012*). The role of Abc1 is unknown and its ability to bind directly to RecBCD has never been tested. Likewise, exactly how Arf influences P22 recombination is unclear, but its small size and acidic primary sequence hint that it might be a DNA mimic protein. Finally, the significance of the interaction between Abc2 and host PpiB is not clear. Our structure shows that the PPI active site almost certainly engages P68 of Abc2, and it is therefore plausible that proline isomerisation may in some way modulate Abc2 activity and therefore P22 infection. In this respect, it is interesting to note that proline isomerisation has been found to be critical for infection of *E. coli* by phage fd (*Eckert et al., 2005*). Further work will be required to fully understand the influence of P22 infection on homologous recombination in bacterial cells which may also have implications of our understanding of RecA loading by the native RecBCD complex.

The RecBCD complex and related systems such as AddAB (also known as RexAB) have been considered as attractive targets for antibacterial drug development (*Lanyon-Hogg, 2021*; *Amundsen et al., 2012*). This reflects the fact that loss of RecBCD/AddAB function reduces infectivity (*Gourley et al., 2017*; *Amundsen et al., 2008*; *Painter et al., 2015*), potentiates the effects of DNA-damaging agents (*Wilkinson et al., 2016b*; *Clarke et al., 2019*; *Clarke et al., 2021*; *Tamae et al., 2008*; *Lim et al., 2019*), and reduces the potential for adaptive mutagenesis leading to antibacterial resistance (*Clarke et al., 2021*; *López et al., 2007*; *Bush et al., 2020*). These effects are all caused by a failure to efficiently respond to DNA breakage caused either by host immune responses or treatment with drugs such as fluoroquinolones. The diverse mechanisms employed by phage to manipulate bacterial DNA break repair may provide inspiration for the ongoing design of small-molecule inhibitors for RecBCD/AddAB (*Lanyon-Hogg, 2021*; *Lim et al., 2019*).

# Materials and methods
## Protein expression and purification
The T7 gp5.9 gene from phage T7 was produced as a synthetic gene construct with an N-terminal 3C-cleavable histidine tag (GeneArt, Invitrogen). This was subcloned into the pACEBAC1 (MultiBac) vector using engineered BamHI and XbaI sites for overexpression in insect cells using standard techniques (*Berger et al., 2004*; *Bieniossek et al., 2008*). Briefly, for large-scale expression of gp5.9, 500 mL of High5 cells at $2 \times 10^6$ cells/mL were infected with 25 mL P3 virus and incubated for 72 hr at 27°C with shaking before cells were harvested by centrifugation. The pellet was resuspended in 50 mL lysis buffer (20 mM Tris-HCl pH 7.5, 200 mM NaCl, 2 mM β-mercaptoethanol, 10% glycerol, protease inhibitor cocktail [Roche, as directed by the manufacturer], 20 mM imidazole). The cells were lysed by sonication and centrifuged to remove cell debris. The supernatant was then applied to Talon resin (Takara Bio) to purify gp5.9 using the histidine tag. Beads were equilibrated by washing three times with 15 mL wash buffer (20 mM Tris-HCl pH 7.5, 200 mM NaCl, 20 mM β-mercaptoethanol, 10% glycerol, 20 mM imidazole). Supernatant from the centrifuged cell lysate was added to the beads and incubated for 30 min at 4°C. The beads were then spun down and the supernatant (unbound protein) was removed. The beads were washed four times with wash buffer before gp5.9 was eluted with 50 mL elution buffer (20 mM Tris-HCl pH 7.5, 200 mM NaCl, 20 mM β-mercaptoethanol, 150 mM imidazole). The eluate was loaded onto a 1 mL MonoQ column (GE Healthcare) in buffer A (20 mM Tris-HCl pH 7.5, 1 mM TCEP) and eluted with buffer B (20 mM Tris-HCl pH 7.5, 1 mM TCEP, 1 M NaCl). Peak fractions were pooled and further purified using size-exclusion chromatography in SEC buffer (20 mM Tris-HCl pH 8, 0.5 mM TCEP, 200 mM NaCl). The concentration of his-tagged gp5.9 was calculated

using a theoretical extinction coefficient of 8480. The final protein preparation was stored at –80°C in a buffer containing 20 mM Tris-HCl-Cl pH 8, 0.5 mM TCEP, 200 mM NaCl at a final concentration of 19.4 µM. A sample of tag-free gp5.9 was prepared by treatment of this stock with 3C protease (Pierce) followed by size-exclusion chromatography, and the resulting protein was stored in the same buffer at a final concentration of 14.9 µM. Unless stated otherwise, all experiments presented in this article were performed with the tag-free gp5.9 protein.

For the wild type RecBCD and nuclease-dead RecB$^D$CD complexes, RecB, RecC, and RecD were co-expressed and purified from three separate plasmids: pETduet-His$_6$-TEVsite-recB or pETduet-His$_6$-TEVsite-recB$^{D1080A}$, pRSFduet-recC and pCDFduet-recD as described previously (*Saikrishnan et al., 2008*; *Wilkinson et al., 2016a*). For the RecBCD-Abc2 complex, the four genes were co-expressed together as Abc2 has been shown to be unstable when overexpressed alone (*Murphy, 1994*). The nuclease-dead RecB mutant (D1080A) was used to prevent digestion of the DNA substrates. To avoid adding a fourth plasmid to the transformation, the genes encoding His$_6$-TEVsite-recB$^{D1080A}$ and RecC were jointly cloned into the respective NdeI/KpnI and NcoI/XhoI sites of the pRSFduet vector for use with the pCDFduet-recD plasmid. Finally, the Abc2 gene from phage P22 was synthesised (GeneStrings, Thermo Fisher) and cloned in-between the NdeI and XhoI sites of pET22b. An Abc2$^{P68A}$ mutant variant was generated using In-Fusion cloning (Takara Bio) for the generation of RecBCD-Abc2 complex without the co-purification of the host PpiB protein. Co-expression of the three plasmid system coding for RecB, RecC, RecD, and Abc2 was performed as for wild-type RecBCD. The RecB-CD-Abc2 complexes were purified by a similar method to that described previously for RecBCD (*Wilkinson et al., 2016a*). Briefly, the cells were lysed using an emulsiflex cell disruptor, the lysate clarified by centrifugation, and ammonium sulphate was added to the soluble fraction (0.35 g/mL) before a second centrifugation at 30,000 × *g* to pellet the proteins. The pellets were resuspended and complexes bound to a HisTrap column (GE Healthcare) before direct elution onto a HiTrap heparin column (GE Healthcare) and subsequent elution using a gradient of NaCl. The His-tags were cleaved by incubation with TEV protease during overnight dialysis before being re-passed through the HisTrap column and finally eluted from a MonoQ column with a shallow gradient of NaCl. The purified proteins were diluted to give a final buffer solution of 25 mM Tris-HCl pH 7.5, 100 mM NaCl, 0.5 mM TCEP. The samples were concentrated to around 5–10 mg/mL, supplemented with glycerol to a concentration of 15% (v/v), and flash-frozen in aliquots in liquid nitrogen for storage at –80°C. The fluorescent SSB biosensor was a gift from Martin Webb (The Crick Institute, London) and was purified as described (*Chisty et al., 2018*).

## Preparation of DNA substrates for cryoEM and native gel shift assays

A splayed hairpin DNA substrate as used in previous cryoEM studies (*Wilkinson et al., 2016a*) was also used here to allow direct comparison of the effect of Abc2 interaction on the conformation of the RecBCD-DNA complex. The oligonucleotide sequence is 5′-TTT TTT TTT TTT tct aat gcg agc act gct aca gca tTT CCC atg ctg tag cag tgc tcg cat tag aTT T-3′, with lowercase denoting paired bases in the duplex region and uppercase denoting unpaired bases (12 on the 5′-end and 3 on the 3′-end). The DNA substrate was purified as described previously (*Saikrishnan et al., 2008*; *Singleton et al., 2004*). Briefly, the synthesised oligonucleotide (IDT) was annealed at low concentration by heating to 95°C followed by a slow cooling to room temperature and purification by chromatography on a Source-Q column. The purified substrate was desalted into deionised water, aliquoted, and stored at –20°C.

## DNA double-strand break resection assays

DSB resection assays were performed in RecBCD buffer (25 mM Tris-HCl pH 7.5, 10 mM NaCl, 6 mM MgCl$_2$, 0.1 mg/mL BSA) supplemented with 2.5 nM RecBCD (or RecBCD-ABC2 or RecBCD-ABC2-PPI) or 20 nM AddAB, 960 µM (in ntds) linearised pACEBac1 vector, and 1 µM gp5.9. Reactions were initiated by the addition of 2 mM ATP. 10 µL aliquots were removed at the times indicated and quenched by the addition of 10 µL STOP buffer (10% glycerol, 1% SDS, 50 mM EDTA, 1 mg/mL proteinase K). The samples were run on 1% agarose gels and imaged by post-staining with SYBR gold.

## Helicase assays

Real-time helicase assays were performed using a fluorescent biosensor for ssDNA based on the *Plasmodium falciparum* SSB protein (fSSB; *Chisty et al., 2018*). Reactions were performed in RecBCD

buffer (25 mM Tris-HCl pH 7.5, 10 mM NaCl, 6 mM MgCl$_2$, 0.1 mg/ml BSA) supplemented with 10 pM RecBCD, 1 uM (in ntds; ~10 pM molecules) bacteriophage lambda DNA (New England Biolabs), 25 nM fSSB (tetramer) and gp5.9 at the stated concentration. After a 10 min pre-incubation, reactions were initiated by the addition of 2 mM ATP. Fluorescence intensity was monitored using a Cary Eclipse Fluorescence Spectrophotometer (excitation wavelength 430 nm, emission wavelength 475 nm, excitation and emission slit widths of 10 nm and 5 nm, respectively). Assays were performed in triplicate, and the initial rates reported are the mean and standard error for the three repeats. To obtain IC$_{50}$ values for inhibition of RecBCD by gp5.9, data describing the initial unwinding rate as a function of $\log_{10}$[gp5.9] were fit to the sigmoidal dose–response equation of GraphPad Prism.

$$\text{rate} = r_{min} + (r_{max} - r_{min})/(1 + 10^{\wedge}(\log IC50 - \log[gp5.9]))$$

$r_{min}$ and $r_{max}$ are the minimum and maximum DNA unwinding rates, respectively. Values for the unwinding rates were normalised to 100% for a zero gp5.9 control, and so for the purposes of the fits the values for $r_{max}$ and $r_{min}$ were constrained to 100 and 0 respectively.

## Native gel mobility shift assays

Native-PAGE was used to assess the shift in mobilities of the RecBCD complex in the presence of DNA and gp5.9. For *Figure 1e*, the 10 µL reactions contained 25 mM Tris-HCl pH 7.5, 10 mM NaCl, 6 mM MgCl$_2$. 50 nM of either RecBCD or RecBCD-Abc2 complex were mixed with 1 µM gp5.9 (or buffer), incubated for 5 min at room temperature and then added to 2.5 nM of a 5′-Cy5-labelled 25 bp duplex DNA substrate (5′-Cy5-GCT TGC TAG GAC GGA TCG CTC GAG G and its complement; IDT). Reactions were then loaded onto a 6% (w/v) native polyacrylamide gel (1× TBE), then run in 1× TBE running buffer for 35 min at a constant voltage of 150 V. The gels were imaged using a Typhoon, scanning for the Cy5 fluorophore. For *Figure 1f*, the 20 µL reactions contained 25 mM Tris-HCl pH 7.5, 10 mM NaCl, 6 mM MgCl$_2$, and 10% glycerol. 400 nM RecBCD or RecBCD-Abc2 were mixed with 4 µM gp5.9 (or buffer) and incubated for 5 min prior to addition of 500 nM of 5′-Cy5-labelled 25 bp duplex DNA substrate. The reactions were incubated for a further 5 min at room temperature prior loading onto a 6% (w/v) native polyacrylamide gel (1× TBE). The gel was run in 1× TBE running buffer for 100 min at a constant voltage of 150 V. The gels were visualised using the Typhoon to scan for the Cy5 fluorophore (to visualise DNA) and then after stained with Coomassie Blue to visualise the proteins.

## CryoEM grid preparation

RecBCD and gp5.9 proteins were thawed and mixed in buffer B100 on ice at final concentrations of 0.3 µM RecBCD, 0.9 µM gp5.9 (based on monomeric weight) and 5 mM MgCl$_2$. Quantifoil-Au 2/1 µm holey carbon film grids (300 mesh) were cleaned by successive washes of MilliQ water and ethylacetate. They were then washed with 0.3 mM n-dodecyl-beta-D-maltoside (DDM) before being immediately covered with a solution of diluted graphene oxide sheets in 0.3 mM DDM, as described previously (*Cheng et al., 2020*). A sample volume of 4 µL was evenly applied to the graphene oxide-coated carbon side of the grid and frozen in liquid ethane using a Vitrobot Mark IV (FEI) with a 10 s wait time and 1.5 s blot time, respectively. The Vitrobot chamber was maintained at close to 100% humidity and 4°C.

RecB$^D$CD-Abc2 was thawed and mixed with a 1.5-fold molar excess of the hairpin DNA substrate for 10 min at room temperature before being placed on ice and ligands and buffer B100 added to give the desired concentrations. The final mixture contained 0.2 µM RecB$^D$CD-Abc2, 0.3 µM DNA, 5 mM magnesium chloride, and 2 mM ADPNP. C-flat 2/1 µm holey carbon film grids (400 mesh) were covered with a solution of diluted graphene oxide sheets in 0.3 mM DDM, as described previously (*Cheng et al., 2020*). The sample (3 µL) was evenly applied to the graphene oxide-coated side of the grid and frozen in liquid ethane using a Vitrobot Mark IV (FEI) with a 2 s wait time and 1.5 s blot time, respectively. The Vitrobot chamber was maintained at close to 100% humidity and 4°C.

## CryoEM data collection

The RecBCD-gp5.9 cryoEM dataset was collected at LonCEM (The Francis Crick Institute, London) using a Titan Krios microscope operated at 300 kV with a Gatan K3 detector in super resolution mode. A nominal magnification of ×81,000 was set yielding a physical pixel size of 1.10 Å used for image processing. A total of 5064 images were collected with a nominal defocus range of –1.0 to –2.5 µm in

0.3 µm increments. Each image consisted of a movie stack of 40 frames with a total dose of ~50 e-/Å$^2$ over 4.3 s, corresponding to a dose rate of ~14 e-/pixel/s (all values relative to the physical pixel size of 1.10 Å).

The RecB$^D$CD-Abc2-DNA-ADPNP cryoEM dataset was collected at eBIC (Diamond Light Source, UK) using a Titan Krios microscope operated at 300 kV with a Gatan K2 detector in counting mode. A nominal magnification of 130,000 was set, yielding a pixel size of 1.06 Å. A total of 2500 images were collected with a nominal defocus range of −1.2 to −2.4 µm in 0.3 µm increments. Each image consisted of a movie stack of 30 frames with a total dose of ~56 e-/Å$^2$ over 12 s, corresponding to a dose rate of ~5.2 e-/pixel/s.

## CryoEM data processing: gp5.9

The 5064 movie stacks were aligned and summed using Motioncor2 (*Zheng et al., 2017*) before CTF parameters were estimated for each micrograph using Gctf (*Zhang, 2016*). Outlying poor quality images, based on CTF figure of merit, defocus value and predicted resolution were removed to give 4080 micrographs for further processing. Template-based particle picking was done with Gautomatch using 25 Å lowpass-filtered (LPF) 2D reprojections of an 8 Å resolution RecBCD-gp5.9 cryoEM map obtained from in-house data on a Technai F20 microscope operating with a Falcon2 detector. A total of 1,458,124 picked particles were extracted 2× binned in RELION3 (*Zivanov et al., 2018*). To remove picking artefacts and noise, two successive rounds of 2D classification were carried out in cryosparc2 (*Punjani et al., 2017*) from which 674,546 potential RecBCD particles were kept (*Figure 2—figure supplement 1b*). A further round of cleaning was carried out using template-free ab initio classification from which two featureless classes were removed to leave 547,993 particles which represented a RecBCD-gp5.9 complex (*Figure 2—figure supplement 1c*). These were refined in cryosparc2, using the ab initio map lowpass filtered to 20 Å as a template, to centre the particles prior to re-extraction without binning in RELION3.

The unbinned, centred particles were refined in RELION3, with the template lowpass filtered to 20 Å and with a soft, extended, 16 Å lowpass filtered mask around the complex, producing a map at a resolution of 3.7 Å (gold-standard, FSC = 0.143, as with all succeeding resolution estimates) after postprocessing. The output particles were then subjected to Bayesian polishing and refined as before in RELION3 producing a map at a resolution of 3.5 Å. A round of CTF refinement of the per-particle defocus values significantly improved the map, which was subsequently refined to a resolution of 3.1 Å (*Figure 2—figure supplement 1d*). A round of 3D classification was then with local angular searches and without a mask to separate any heterogeneity within the complex into three classes (*Figure 2—figure supplement 1e*). A class containing 163,484 particles (30% of those classified) with strong density for the full complex was selected out from low-resolution classes with mixed occupancies and refined to a resolution of 3.2 Å (*Figure 2—figure supplement 1f*). A focused 3D classification was then run using a soft mask around the gp5.9 and RecB arm domain region of the complex and without image alignment. This led to the exclusion of 22,026 particles with no gp5.9 density and selection of 141,458 particles with strong gp5.9 density (*Figure 2—figure supplement 1g*), which were refined to yield a final map at 3.2 Å (*Figure 2—figure supplement 1h*), which was deposited with a sharpening factor of −50 e-/Å$^2$ applied. Due to the reduced resolution still observed around the C-terminus of gp5.9 and the RecB arm in the map, the refinement was continued using the local mask from the previous classification to generate a secondary map to aid model building and interpretation in this region.

## CryoEM data processing: Abc2

The movie stacks were aligned and summed using Motioncor2 (*Zheng et al., 2017*) before CTF parameters were estimated for each micrograph using Gctf (*Zhang, 2016*). Images with significant ice contamination were removed to give 2338 micrographs for further processing. Template-based particle picking was done with Gautomatch using 25 Å LPF 2D-reprojections of the published RecBCD-DNA-ADPNP cryoEM map (EMD-4038, *Wilkinson et al., 2016a*). A total of 467,613 picked particles were extracted 2× binned. To remove picking artefacts and noise, two successive rounds of 2D classification were carried out in RELION3 (*Zivanov et al., 2018*) from which 185,881 potential RecBCD particles were kept (*Figure 4—figure supplement 1b*). These were used for an unmasked 3D refinement with the EMD-4038 map filtered to 30 Å resolution as the starting template to centre

the particles before re-extraction without binning. This was followed by 3D refinement with a soft mask around the full complex, Bayesian polishing and CTF refinement of the per-particle defocus values. The map resolution improved from 3.6 Å (0.143 FSC cutoff, RELION3) to 3.3 Å after Bayesian polishing, CTF refinement, and a further 2D classification without image alignment to select 167,080 highly ordered particles (*Figure 4—figure supplement 1c and d*).

To separate different states/occupancies of the complex, a focused refinement was done on the polished particles with a soft mask around Abc2 and the surrounding domains of RecC, followed by 3D classification without image alignment using the same focused mask (*Figure 4—figure supplement 1e and f*). The majority of the particles (71% of those classified) formed a high-resolution class with strong density for the full complex, including Abc2 bound on the surface of RecC. A minor, low-resolution class was excluded containing 7% of the particles respectively. The remaining particles (22% of those classified) were in a class that displayed additional density attached to Abc2 for the contaminant *E. coli* PpiB protein, representing the RecBCD-DNA-Abc2-PPI complex. The 119,163 RecBCD-DNA-Abc2 particles were refined to generate a map at a resolution of 3.4 Å (0.143 FSC cutoff, RELION3) deposited with a sharpening B-factor of –50 Å$^2$ (*Figure 4—figure supplement 1g and i*, *Table 2*). The 37,072 RecBCD-DNA-Abc2-PPI particles were refined to yield a map at a resolution of 3.8 Å (0.143 FSC cutoff, RELION3), which was deposited with a sharpening B-factor of –50 Å$^2$ (*Figure 4—figure supplement 1h and i*, *Table 2*).

## Model building and refinement

For the 3.2 Å resolution RecBCD-gp5.9 map, RecBCD from the published RecBCD-GamS structure (PDB ID: 5MBV, *Anderson and Kowalczykowski, 1997b*) was docked into the density using Chimera (*Dillingham et al., 2008*) and refined with jelly-body restraints using Refmac5 (*Brown et al., 2015*) in CCPEM (*Burnley et al., 2017*). A focused refinement in RELION3 with the mask around the gp5.9 binding site used during data processing (*Figure 2—figure supplement 1g*) generated a more resolved map to facilitate the building of gp5.9 and remodelling of interacting regions of the RecB arm and RecC C-terminal domains respectively in Coot (*Konno et al., 1996*). Initially a long idealised α-helix was docked in to start gp5.9 building with an N-terminal ß-strand extension clearly visible. At this resolution and with the majority of the peptide ordered, sequence assignment was clear and straightforward with residues 1–49 modelled. The chain was duplicated and docked into the density for the second gp5.9 molecule within the homo-dimer, with only minor adjustments required for the very N-terminal residues and some altered side-chain conformers (*Figure 2—figure supplement 2*). An AlphaFold (*Yüksel et al., 2016*) model of gp5.9 was later used to help validate our model. Notably, an AlphaFold model of the RecBCD complex combined with observations of inconsistencies between the model and map in the RecB arm led to the finding that the protein register within this domain was off by seven residues for residues 241–289, which are flanked by disordered loops in prior RecBCD structures. Two tryptophan residues positioned seven residues apart in this region had become the basis for the misalignment in the first model built for RecBCD (*Singleton et al., 2004*), and it was not uncovered with the ambiguous nature of the density in this region in the subsequent RecBCD structures that followed, until this significantly higher-resolution gp5.9 complex map (*Figure 2—figure supplement 4*). We went back through past RecBCD complex structures and found that the new RecB arm model fit significantly better into all of the past maps. Comparative analyses of RecBCD structures in this article use the corrected models.

The published RecBCD-DNA-ADPNP structure (PDB ID: 5LD2, *Wilkinson et al., 2016a*) with the corrected RecB arm domain was docked into the 3.4 Å Abc2-containing map using Chimera and refined with jelly-body restraints using Refmac5 (*Brown et al., 2015*) in CCPEM. Additional density was present on the surface of RecC that was sufficiently ordered and resolved to facilitate manual building of the novel Abc2 polypeptide chain from residues 6–52 (*Figure 4—figure supplement 4a*) in Coot with iterative rounds of manual building and Phenix real-space refinement (*Adams et al., 2010*). Initially, three α-helices were docked in and connecting loops built. Sequence was assigned based on a combination of secondary structure prediction generated by JPRED (*Drozdetskiy et al., 2015*) and register matching of density size and environment to the chemical properties of the known amino acid sequence. Concurrently, the surrounding RecC region was adjusted to fit the density with minor remodelling except for a surface bundle including residues 252–294 (*Figure 4—figure supplement 3b*). This helical bundle peels back to allow Abc2 binding and then loosely clamps back to hold

**Table 2.** CryoEM data collection, refinement, and validation statistics.

| | gp5.9-RecBCD (EMDB-15803) (PDB 8B1R) | Abc2-RecBCD-DNA-ADPNP (EMDB-15804) (PDB 8B1T) | Abc2-RecBCD-PpiB-DNA-ADPNP (EMDB-15805) (PDB 8B1U) |
|---|---|---|---|
| **Data collection and processing** | | | |
| Magnification | 81,000 | 130,000 | |
| Voltage (kV) | 300 | 300 | |
| Detector | K3 | K2 | |
| Energy slit width (e⁻V) | 20 | 20 | |
| Electron exposure ($e^-/Å^2$) | 50 | 56 | |
| Exposure rate (e⁻/pixel/s) | 14.0 | 5.2 | |
| Defocus range (μm) | −1.0 to −2.5 | −1.2 to −2.4 | |
| Pixel size (Å) | 1.10 | 1.06 | |
| Movies collected | 5064 | 2500 | |
| Initial particle images (no.) | 674,546 | 185,881 | |
| Final particle images (no.) | 141,458 | 119,163 | 37,072 |
| Symmetry imposed | C1 | C1 | C1 |
| Map resolution (Å) | 3.2 | 3.4 | 3.8 |
| FSC threshold | 0.143 | 0.143 | 0.143 |
| Map resolution range (Å) | | | |
| | | | |
| **Refinement** | | | |
| Initial model used (PDB code) | 5MBV | 5LD2, 8B1R | 8B1T |
| Map sharpening $B$ factor ($Å^2$) | −50 | −50 | −50 |
| Model resolution (Å) | 3.1 | 3.3 | 3.7 |
| FSC threshold | 0.143 | 0.143 | 0.143 |
| Model to map correlation | 0.87 | 0.83 | 0.78 |
| **Model composition** | | | |
| Non-hydrogen atoms | 21,953 | 23,986 | 24,127 |
| Protein residues | 2746 | 2869 | 2886 |
| DNA residues | - | 51 | 51 |
| Ligand molecules | 1 ($Mg^{2+}$) | 1 (ADPNP) 1 ($Mg^{2+}$) | 1 (ADPNP) 1 ($Mg^{2+}$) |
| **$B$ factors ($Å^2$)** | | | |
| Protein | 89.5 | 72.0 | 68.0 |
| DNA | - | 168.5 | 162.6 |
| Ligand | 61.0 | 50.0 | 54.5 |
| **R.m.s. deviations** | | | |
| Bond lengths (Å) | 0.004 | 0.003 | 0.002 |
| Bond angles (°) | 0.617 | 0.493 | 0.464 |

*Table 2 continued on next page*

*Table 2 continued*

| | gp5.9-RecBCD (EMDB-15803) (PDB 8B1R) | Abc2-RecBCD-DNA-ADPNP (EMDB-15804) (PDB 8B1T) | Abc2-RecBCD-PpiB-DNA-ADPNP (EMDB-15805) (PDB 8B1U) |
|---|---|---|---|
| Validation | | | |
| MolProbity score | 1.4 | 1.5 | 1.5 |
| Clashscore | 7.5 | 6.0 | 5.7 |
| Poor rotamers (%) | 0.5 | 1.4 | 1.5 |
| Ramachandran plot | | | |
| Favoured (%) | 99.3 | 98.9 | 98.9 |
| Allowed (%) | 0.7 | 1.1 | 1.1 |
| Disallowed (%) | 0.0 | 0.0 | 0.0 |

Abc2 in place, although the density in its new position was insufficient for model building (*Figure 4—figure supplement 3c*).

For the Abc2 and PPI-containing map at 3.8 Å resolution, the final RecBCD-DNA-Abc2 model was docked in and refined with jelly-body restraints as described above. As well as containing additional density for PPI, the map showed further density for building an extended part of the C-terminal region of the Abc2 model up to residue 66 (*Figure 4—figure supplements 4b and 5a*). The density in this region was ambiguous as indicated by local resolution estimation identifying the resolution to be worse than 5 Å for PpiB (*Figure 4—figure supplement 1h*). As a result, the crystal structure of *E. coli* PpiB (PDB: 1LOP) was docked into the density (*Figure 4—figure supplement 4*) for the purpose of figures but not for the deposition of a 3D model. It is unclear what changes are induced in Abc2 and in PPI itself upon PPI binding to the C-terminal portion of Abc2. However, the model is strongly suggestive of P68 of Abc2 binding within the active site of the prolyl isomerase (*Figure 4f*). The final models of each structure were real-space refined in Phenix against the deposited maps with ADP-factor refinement enabled and model quality assessed by MolProbity (*Williams et al., 2018*) to generate the final model statistics (*Table 2*).

## Materials availability statement

Expression vectors used in this study are available from the corresponding authors upon request.

## Data availability statement

Newly generated structural models from this work have been deposited at the PDB and EMDB. Accession codes are shown in *Table 2*. Existing structural models used in this work are accessible from the PDB with accession codes 5MBV and 5LD2.

## Acknowledgements

Work in the MSD laboratory was funded by the Wellcome Trust (100401/Z/12/Z) and the BBSRC (BB/S007261/1). Work in the DBW laboratory was funded by Cancer Research UK (C6913/A2160), the Wellcome Trust (209327/Z/17/Z), and the MRC (MR/N009258/1). We thank Martin Webb (The Crick Institute) for the gift of the SSB biosensor and Emma Galletti di Cadilhac (Bristol) for technical assistance.

## Additional information

### Funding

| Funder | Grant reference number | Author |
|---|---|---|
| Wellcome Trust | 100401/Z/12/Z | Mark S Dillingham |
| Biotechnology and Biological Sciences Research Council | BB/S007261/1 | Mark S Dillingham |
| Cancer Research UK | C6913/A2160 | Dale B Wigley |
| Wellcome Trust | 209327/Z/17/Z | Dale B Wigley |
| Medical Research Council | MR/N009258/1 | Dale B Wigley |

The funders had no role in study design, data collection and interpretation, or the decision to submit the work for publication. For the purpose of Open Access, the authors have applied a CC BY public copyright license to any Author Accepted Manuscript version arising from this submission.

### Author contributions

Martin Wilkinson, Conceptualization, Data curation, Formal analysis, Validation, Investigation, Methodology, Writing – original draft, Writing – review and editing; Oliver J Wilkinson, Formal analysis, Investigation, Methodology, Writing – review and editing; Connie Feyerherm, Emma E Fletcher, Formal analysis, Investigation; Dale B Wigley, Conceptualization, Supervision, Funding acquisition, Writing – review and editing; Mark S Dillingham, Conceptualization, Formal analysis, Supervision, Funding acquisition, Writing – original draft, Writing – review and editing

### Author ORCIDs

Martin Wilkinson ⓘ http://orcid.org/0000-0001-5490-613X
Oliver J Wilkinson ⓘ http://orcid.org/0000-0003-4107-6434
Dale B Wigley ⓘ http://orcid.org/0000-0002-0786-6726
Mark S Dillingham ⓘ http://orcid.org/0000-0002-4612-7141

### Decision letter and Author response

Decision letter https://doi.org/10.7554/eLife.83409.sa1
Author response https://doi.org/10.7554/eLife.83409.sa2

## Additional files

### Supplementary files
• MDAR checklist

### Data availability

Source data files have been provided for gel-based analyses (Figure 1 - Source data 1). All new cryoEM data/models generated in this work have been deposited at the EMDB and PDB under accession codes EMDB-15803, EMDB-15804, EMDB-15805, PDB 8B1R, PDB 8B1T and PDB 8B1U. Validation reports have been provided for structural models with submission.

The following datasets were generated:

| Author(s) | Year | Dataset title | Dataset URL | Database and Identifier |
|---|---|---|---|---|
| Wilkinson M, Wilkinson OJ, Feyerherm C, Fletcher EE, Wigley DB, Dillingham MS | 2022 | RecBCD in complex with the phage protein gp5.9 | https://www.rcsb.org/structure/8B1R | RCSB Protein Data Bank, 8B1R |

*Continued on next page*

*Continued*

| Author(s) | Year | Dataset title | Dataset URL | Database and Identifier |
|---|---|---|---|---|
| Wilkinson M, Wilkinson OJ, Feyerherm C, Fletcher EE, Wigley DB, Dillingham MS | 2022 | RecBCD-DNA in complex with the phage protein Abc2 | https://www.rcsb.org/structure/8B1T | RCSB Protein Data Bank, 8B1T |
| Wilkinson M, Wilkinson OJ, Feyerherm C, Fletcher EE, Wigley DB, Dillingham MS | 2022 | RecBCD-DNA in complex with the phage protein Abc2 and host PpiB | https://www.rcsb.org/structure/8B1U | RCSB Protein Data Bank, 8B1U |
| Wilkinson M, Wilkinson OJ, Feyerherm C, Fletcher EE, Wigley DB, Dillingham MS | 2022 | RecBCD in complex with the phage protein gp5.9 | https://www.ebi.ac.uk/emdb/EMD-15803 | EMDB, EMD-15803 |
| Wilkinson M, Wilkinson OJ, Feyerherm C, Fletcher EE, Wigley DB, Dillingham MS | 2022 | RecBCD-DNA in complex with the phage protein Abc2 | https://www.ebi.ac.uk/emdb/EMD-15804 | EMDB, EMD-15804 |
| Wilkinson M, Wilkinson OJ, Feyerherm C, Fletcher EE, Wigley DB, Dillingham MS | 2022 | RecBCD-DNA in complex with the phage protein Abc2 and host PpiB | https://www.ebi.ac.uk/emdb/EMD-15805 | EMDB, EMD-15805 |

The following previously published datasets were used:

| Author(s) | Year | Dataset title | Dataset URL | Database and Identifier |
|---|---|---|---|---|
| Wilkinson M, Chaban Y, Wigley DB | 2017 | Cryo-EM structure of Lambda Phage protein GamS bound to RecBCD. | https://www.rcsb.org/structure/5MBV | RCSB Protein Data Bank, 5MBV |
| Wilkinson M, Chaban Y, Wigley DB | 2016 | Cryo-EM structure of RecBCD+DNA complex revealing activated nuclease domain | https://www.rcsb.org/structure/5LD2 | RCSB Protein Data Bank, 5LD2 |

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
