## [Editor Report]

This important study addresses the ways in which bacteriophages antagonise or coopt the DNA restriction and/or recombination functions of the bacterial RecBCD helicase-nuclease. The evidence from both biochemistry and structural biology showing convergent evolution is convincing.

---

## [Decision Letter]

**Decision letter after peer review:**

Thank you for submitting your article "Structures of RecBCD in complex with phage-encoded inhibitor proteins reveal distinctive strategies for evasion of a bacterial immunity hub." for consideration by *eLife*. Your article has been reviewed by 3 peer reviewers, one of whom is a member of our Board of Reviewing Editors, and the evaluation has been overseen Volker Dötsch as the Senior Editor. The reviewers have opted to remain anonymous.

Essential revisions:

1) Interestingly, Abc2 cannot be purified by itself alone but is stable only in complexes with RecBCD. Because of a Proline residue (Pro68) in Abc2, which is a substrate of prolyl-isomerase (PPI), WT Abc2 is tightly associated with PPI, but the mutant Abc2P68A can be separated from PPI. Therefore, the authors have prepared both RecBCD- Abc2P68A and RecBCD- Abc2-PPI. The biochemical characterization of the effects of Abc2 on RecBCD is a repeat of KC Murphy's paper, but different from KC Murphy's in the effects of Abc2 on dsDNA-end binding (2-4 fold increase, by Murphy) and helicase activity (3-4 fold reduced, by Murphy) of RecBCD (reference 15). Here, both RecBCD- Abc2P68A and RecBCD- Abc2-PPI have comparable enzymatic activities as RecBCD alone and both can be blocked by gp5.9 as by Gam (Murphy). The cryoEM structures reveal Abc2 binds the Chi-recognition RecC subunit and potentially modifies RecBCD in response to the Chi sequence. But in the absence of DNA, the structure does not explain the in vivo function of Abc2 hijacking RecBCD, nor how Abc2 alters dsDNA binding and helicase activity of RecBCD as reported by Murphy. This needs to be discussed.

2) The manuscript seems to be written primarily for RecBCD experts. Please explain RecBCD function and mechanism in DSB repair (Chi sequence and RecA recruitment) and in CRISPR activity better and more clearly. For example, the diagram in Figure 1b shows the exonuclease activity of RecBCD, but how could the single nucleotides of phage DNA be incorporated into CRISPR? Even with the endonuclease activity, RecBCD produces single-stranded phage DNA fragments only. How this ssDNA be incorporated into CRISPR is unclear.

3) Whereas the gp5.9 structure fully rationalizes the effect of gp5.9 on RecBCD activity, the Abc2 structure – while illuminating the docking site on RecBCD, a clear advance – does not clarify how Abc2 impacts RecBCD function. The authors speculate that Abc2 binding prevents RecA loading on the unwound DNA 3' strand while favoring the loading of the phage recombinase Erf.

Does the structure provide impetus and clues for further experiments to elaborate on that question and, if so, how?

*Reviewer #2 (Recommendations for the authors):*

1) Please be consistent and clear in Methods in defining buffers as Tris-HCl (if this is indeed what they are) rather than "Tris" or "tris-Cl".

2) There should be spaces between numbers and units: e.g., please correct "pH7.5" and "20nM" on p. 7.

3) Be consistent in the use of "min" as a standard abbreviation to signify minutes. Please correct "mins" on p. 8.

4) Figure S1 legend. Please specify how the polypeptides were stained. (Coomassie Blue?)

*Reviewer #3 (Recommendations for the authors):*

The authors may improve the manuscript by clarifying the following points.

1. The manuscript is written for the RecBCD experts only. It is rather cryptic to all others. The authors may explain the RecBCD function and mechanism in DSB repair (Chi sequence and RecA recruitment) and in CRISPR better and more clearly. For example, the diagram in Figure 1b shows the exonuclease activity of RecBCD, but how could the single nucleotides of phage DNA be incorporated into CRISPR? Even with the endonuclease activity, RecBCD produces single-stranded phage DNA fragments only. How this ssDNA is incorporated into CRISPR is unclear to this reader.

2. Is phage-encoded Abc2 not species specific? Can *Salmonella* phage P22 infect *E. coli*?

3. Could the authors discuss and explain the different results of Abc2-RecBCD between their group and Murphy's, such as DNA substrate and assay setup?

4. Is gp5.9 alone a dimer and folded as in the complex of RecBCD-go5.9?

5. When describing the RecBCD-gp5.9 structure, additional details may be included. For Example, which residues in gp5.9 are involved in specific interactions with RecBCD and cannot bind AddAB?

6. In Figure 3, additions of residue numbers of some side chains will help orient readers. Based on the blue (involving binding) and grey colors (not involving binding), Many more RecB residues are in contact with DNA than interacting with gp5.9 or Gam inhibitors. Could the authors please explain how gp5.9 can inhibit RecBCD from DNA binding?

---

## [Author Response]

Essential revisions:1) Interestingly, Abc2 cannot be purified by itself alone but is stable only in complexes with RecBCD. Because of a Proline residue (Pro68) in Abc2, which is a substrate of prolyl-isomerase (PPI), WT Abc2 is tightly associated with PPI, but the mutant Abc2P68A can be separated from PPI. Therefore, the authors have prepared both RecBCD- Abc2P68A and RecBCD- Abc2-PPI. The biochemical characterization of the effects of Abc2 on RecBCD is a repeat of KC Murphy's paper, but different from KC Murphy's in the effects of Abc2 on dsDNA-end binding (2-4 fold increase, by Murphy) and helicase activity (3-4 fold reduced, by Murphy) of RecBCD (reference 15). Here, both RecBCD- Abc2P68A and RecBCD- Abc2-PPI have comparable enzymatic activities as RecBCD alone and both can be blocked by gp5.9 as by Gam (Murphy). The cryoEM structures reveal Abc2 binds the Chi-recognition RecC subunit and potentially modifies RecBCD in response to the Chi sequence. But in the absence of DNA, the structure does not explain the in vivo function of Abc2 hijacking RecBCD, nor how Abc2 alters dsDNA binding and helicase activity of RecBCD as reported by Murphy. This needs to be discussed.

Regarding the first point (Murphy’s results). We have now included more detail about Murphy’s results and a brief comparative discussion of our own (page 13). An important caveat in interpreting small (<5-fold) effects on RecBCD activity is that the complex is known to possess different levels of specific activity between preparations (from 20% to 100% active based on titration of DNA ends). This is especially problematic when assessing the effect of Abc2 on RecBCD because (unlike gp5.9 for instance) the protein cannot be purified in isolation and titrated into free RecBCD to monitor how activity changes. Instead, one is comparing activity between different preparations either including Abc2 or not.

Regarding the second point (how much does the structure tells us about the mechanism of Abc2?). We agree with the general sentiment here: the mechanism of RecBCD hijacking by Abc2 is still a “work in progress”. Nevertheless, the structure is suggestive of effects on Chi recognition and/or RecA loading which is both consistent with biochemical results and suggests new avenues for further investigation. This is highly related to point (3) and so we discuss it further below in that response.

2) The manuscript seems to be written primarily for RecBCD experts. Please explain RecBCD function and mechanism in DSB repair (Chi sequence and RecA recruitment) and in CRISPR activity better and more clearly. For example, the diagram in Figure 1b shows the exonuclease activity of RecBCD, but how could the single nucleotides of phage DNA be incorporated into CRISPR? Even with the endonuclease activity, RecBCD produces single-stranded phage DNA fragments only. How this ssDNA be incorporated into CRISPR is unclear.

We have added more detail to the introduction to better explain RecBCD function (page 3). Note that the mechanism by which RecBCD initiates CRISPR adaption is currently unknown and involves additional factors including Cas1 and Cas2. What does seem to be clear is that DNA sequences upstream of Chi somehow become integrated into CRISPR arrays in a RecBCD-dependent manner. This strongly suggests that the CRISPR adaption substrates include degradation products from RecBCD end-processing reactions.

RecBCD does not typically produce single nucleotides as products but rather stretches of ssDNA which vary greatly in length depending on the solution conditions (especially the ATP/Mg^2+^ ratio). We have now altered both the text and figures in the paper to hopefully avoid giving the impression that the products of RecBCD degradation are single nucleotides (Figure 5 and legend).

3) Whereas the gp5.9 structure fully rationalizes the effect of gp5.9 on RecBCD activity, the Abc2 structure – while illuminating the docking site on RecBCD, a clear advance – does not clarify how Abc2 impacts RecBCD function. The authors speculate that Abc2 binding prevents RecA loading on the unwound DNA 3' strand while favoring the loading of the phage recombinase Erf.Does the structure provide impetus and clues for further experiments to elaborate on that question and, if so, how?

We accept this point. While the RecBCD-gp5.9 structure “nails” the inhibition mechanism as steric exclusion of substrate, the RecBCD-Abc2 structure is less informative. Previously published biochemical and in vivo analyses of Abc2 show that it modulates rather than completely inhibits the enzyme. The hypothesis is that Abc2 modifies the process of Chi recognition and/or RecA loading (which are themselves coupled processes) in order to facilitate loading of the phage recombinase Erf. Given current structural models for the mechanism of RecBCD, it is not entirely obvious from the structure of RecBCD-Abc2 what exactly this small phage protein is doing, because (a) there is no significant change to the structure of RecBCD induced by Abc2 interaction and (b) no *known* protein interaction site (eg with RecA) is blocked. Indeed, our original manuscript ended with an acknowledgement that understanding how P22 controls recombination in *E. coli* was ongoing work.

As we see it, in addition to simply revealing the binding site of Abc2, our structure has two significant impacts. Firstly, it is consistent with and extends the existing hypothesis. For example, (a) the interaction of Abc2 with RecC is precisely with the domains of the protein that are responsible for Chi recognition and close to a *putative* site of RecA loading; (b) the recognition that a conserved proline in Abc2 interacts with the active site of PPI implies that Abc2 function is dependent on proline isomerisation; (c) the possible bridging of RecB and RecC by the C-and N-terminal regions of the protein suggest that Abc2 might hinder inter-subunit conformational changes. Secondly, the structure facilitates the testing of this hypothesis. For example, (a) does RecA and/or Erf loading depend on interactions with the surfaces destroyed or created by Abc2 at the interface with RecC (b) does P68A mutation inactivate Abc2?; (c) does failure to recognise and respond to Chi require bridging of RecB and RecC that limits conformational transitions? Crucially, as we explain in the discussion, the future study of the P22 recombination system will require the purification and characterisation of additional factors (Abc2, Arf and Erf) beyond just Abc2. This is something we are working on currently in the lab and consider to be beyond the scope of this work.

Reviewer #2 (Recommendations for the authors):1) Please be consistent and clear in Methods in defining buffers as Tris-HCl (if this is indeed what they are) rather than "Tris" or "tris-Cl".

We accept the point and have modified the methods accordingly throughout.

2) There should be spaces between numbers and units: e.g., please correct "pH7.5" and "20nM" on p. 7.

We have changed the text as requested (pg 17).

3) Be consistent in the use of "min" as a standard abbreviation to signify minutes. Please correct "mins" on p. 8.

We have changed the text as requested (pg 18).

4) Figure S1 legend. Please specify how the polypeptides were stained. (Coomassie Blue?)

Yes, the SDS-PAGE gels were stained with Coomassie. We have altered the figure legend as requested (pg 42).

Reviewer #3 (Recommendations for the authors):The authors may improve the manuscript by clarifying the following points.1. The manuscript is written for the RecBCD experts only. It is rather cryptic to all others. The authors may explain the RecBCD function and mechanism in DSB repair (Chi sequence and RecA recruitment) and in CRISPR better and more clearly. For example, the diagram in Figure 1b shows the exonuclease activity of RecBCD, but how could the single nucleotides of phage DNA be incorporated into CRISPR? Even with the endonuclease activity, RecBCD produces single-stranded phage DNA fragments only. How this ssDNA is incorporated into CRISPR is unclear to this reader.

See essential revisions section point (2).

2. Is phage-encoded Abc2 not species specific? Can Salmonella phage P22 infect *E. coli*?

A literature search suggests that the P22 phage cannot infect *E. coli* (at least the K and C strains). However, expression of P22 Abc2 protein alone in *E. coli* confers phenotypes which are associated with *recb*/*recc* knockout such as ciprofloxacin sensitivity (our unpublished work). This strongly suggests that Abc2 is active against RecBCD in vivo in a cross-species manner. This is perhaps unsurprising given the very high sequence similarity between *E. coli* and *Salmonella* RecC proteins (~89% identity). Moreover, P22-like coliphage (eg HK542) encode Abc2 proteins with >90% identity to the P22 protein. Nevertheless, we thank the reviewer for making this interesting point because it remains plausible that the effects of Abc2 on the *E. coli* enzyme may not fully recapitulate those on its natural target.

3. Could the authors discuss and explain the different results of Abc2-RecBCD between their group and Murphy's, such as DNA substrate and assay setup?

See essential revisions section points (1) and (3).

4. Is gp5.9 alone a dimer and folded as in the complex of RecBCD-go5.9?

We don’t have any data on this. However, gp5.9 displays the typical coiled-coil pattern of conserved leucines which pack to form a hydrophobic core between two helices, so we would expect the answer to be ‘yes’.

5. When describing the RecBCD-gp5.9 structure, additional details may be included. For Example, which residues in gp5.9 are involved in specific interactions with RecBCD and cannot bind AddAB?

We have added more detail to our analysis of the RecBCD-gp5.9 structure to address this point. Specifically, we now include a complete table of contacts formed between RecBCD and gp5.9 as Source Data for Figure 3 which includes all H-bonding contacts at the interface rather than just ion-pair contacts (which were emphasised to illustrate DNA mimicry). We also highlight this new information in the main text. For completeness and consistency, we also made an equivalent Supplementary Table for the Abc2 structure (Source data Figure 4)

6. In Figure 3, additions of residue numbers of some side chains will help orient readers. Based on the blue (involving binding) and grey colors (not involving binding), Many more RecB residues are in contact with DNA than interacting with gp5.9 or Gam inhibitors. Could the authors please explain how gp5.9 can inhibit RecBCD from DNA binding?

We have added residue numbers for the interacting side chains as requested (Figure 3 and accompanying legend). Regarding the second point, the structure shows that gp5.9 inhibits RecBCD from binding DNA by steric exclusion. The relative binding affinities of DNA and gp5.9 for this site, as well as the concentrations of DNA breaks and gp5.9 in the cell, will determine the efficacy of gp5.9 as an inhibitor. Note that interactions between RecBCD and gp5.9 are more extensive than just the ion pair contacts shown in Figure 3 (see new Supplementary Tables and the response to point 5). Empirically, inhibition of RecBCD by gp5.9 is quite potent in vitro (IC_50_ = 1.5 nM under our experimental conditions).